# Machine learning using rapidity-mass matrices for event classification problems in HEP

S. V. Chekanov

*HEP Division, Argonne National Laboratory,*

*9700 S. Cass Avenue, Argonne, IL 60439, USA*

(Dated: October 9, 2018)

## Abstract

Supervised artificial neural networks (ANN) with the rapidity-mass matrix (RMM) inputs were studied using several Monte Carlo event samples for various $pp$ collision processes. The study shows the usability of this approach for general event classification problems. The proposed standardization of the ANN feature space can simplify searches for signatures of new physics at the LHC when using machine learning techniques. In particular, we illustrate how to improve signal-over-background ratios in searches for new physics, how to filter out Standard Model events for model-agnostic searches, and how to separate gluon and quark jets for Standard Model measurements.

## I.  INTRODUCTION

Transformations of lists with four-momenta of particles produced in collision events into the rapidity-mass matrices (RMM) [1] that encapsulate information on single- and two-particle densities of identified particles and jets can lead to a systematic approach for defining input variables for various artificial neural networks (ANNs) used in particle physics. By construction, the RMMs are expected to be sensitive to a wide range of popular event signatures of the Standard Model (SM), and thus can be used for various searches of new signatures Beyond the Standard Model (BSM). We remind that the diagonal elements of RMM represent transverse momenta of all objects, the upper-right elements are invariant masses of each two-particle combination, while the lower-left cells reflect rapidity differences. Event signatures with missing transverse energies and Lorentz factors are also conveniently included. The usefulness of the RMM formalism has been illustrated in [1] using a simple example of background reduction for charged Higgs searches.

In the past, separate variables of the RMM have already been used for the "feature" space for machine learning applications in particle collisions. A recent example of a machine learning that uses the numbers of jets, jet transverse momenta, and rapidities as inputs for neural network algorithms can be found in [2]. Unlike a handcrafted set of variables, the RMM represents a well-defined organization principle for creating unique "fingerprint" of collision events suitable for a large variety of event types and ANN architectures due to the unambiguous mapping of a broad number of experimental signatures to the ANN nodes. Therefore, a time-consuming determination of feature space for every physics topics, as well as preparations of this feature space (i.e. re-scaling, normalization, de-correlation etc.) for machine learning may not be required since RMMs already satisfy the most standard requirements for supervised machine learning algorithms.

The results presented in this paper confirm that the standard RMM transformation is a convenient choice for general event classification problems using supervised machine learning. In particular, we will illustrate that repetitive and tedious tasks of feature-space engineering to identify ANN inputs for different event categories can be fully or partially automated. This paper illustrates a few use cases of this technique. In particular, we show how to improve signal-over-background ratios in searches for BSM physics (Sect. III), how to filter out SM events for model-agnostic searches (Sect. IV), and how to separate gluon and quark

jets for SM measurements (Sect. V).

## II. EVENT CLASSIFICATION WITH RMM

In this section, we will illustrate that the feature space in the form of the standard RMM can conveniently be applied for event-classification problems for a broad class of $pp$ collision processes simulated with the help of Monte Carlo (MC) event generators.

This analysis is based on the Pythia8 MC model [3] for generation of $pp$-collision events at $\sqrt{s} = 13$ TeV centre of mass energy. The NNPDF 2.3 LO [4] parton density function from the LHAPDF library [5] was used. The following five collision processes were simulated: (1) multijet QCD events, (2) Standard Model (SM) Higgs production, (3) $t\bar{t}$ production, (4) double-boson production and (5) charged Higgs boson ($H^+t$) process using the diagram $bg \rightarrow H^+t$ for models with two (or more) Higgs doublets [6]. A minimum value of 50 GeV on generated invariant masses of the $2 \rightarrow 2$ system was set. For each event category, all available sub-processes were simulated at leading-order QCD with parton showers and hadronization. Stable particles with a lifetime of more than $3 \cdot 10^{-10}$ seconds were considered, while neutrinos were excluded from consideration. All decays of top quarks, $H$ and vector bosons were allowed. The files with the events were archived in the HepSim repository [7].

Jets, isolated electrons, muons and photons were reconstructed using the procedure described in [1]. Jets were constructed with the anti-$k_T$ algorithm [8] as implemented in the FASTJET package [9] using a distance parameter of $R = 0.4$. The minimum transverse energy of all jets was 40 GeV in the pseudorapidity range of $|\eta| < 2.5$. Jets were classified as light-flavor and as $b$-jets, which were identified by matching the momenta of $b$-quarks with reconstructed jets, and requiring that the total momenta of $b$-quarks should be larger than 50% of the total jet energy. The $b$-jet fake rate was also included assuming that it increases from 1% to 6% at the largest $p_T$ [10].

Muons, electrons and photons were reconstructed from Pythia8 truth-level information after applying isolation criteria [1]. These particles were reconstructed after applying an isolation radius of 0.1 in the $\eta - \phi$ space around the lepton direction. A lepton is considered to be isolated if it carries more than 90% of the cone energy. To simulate the electron fake rate, we replaced jets with the number of constituents less than 10 with the electron ID using 10% probability. In the case of muons, we used a 1% misidentification rate, i.e. replacing

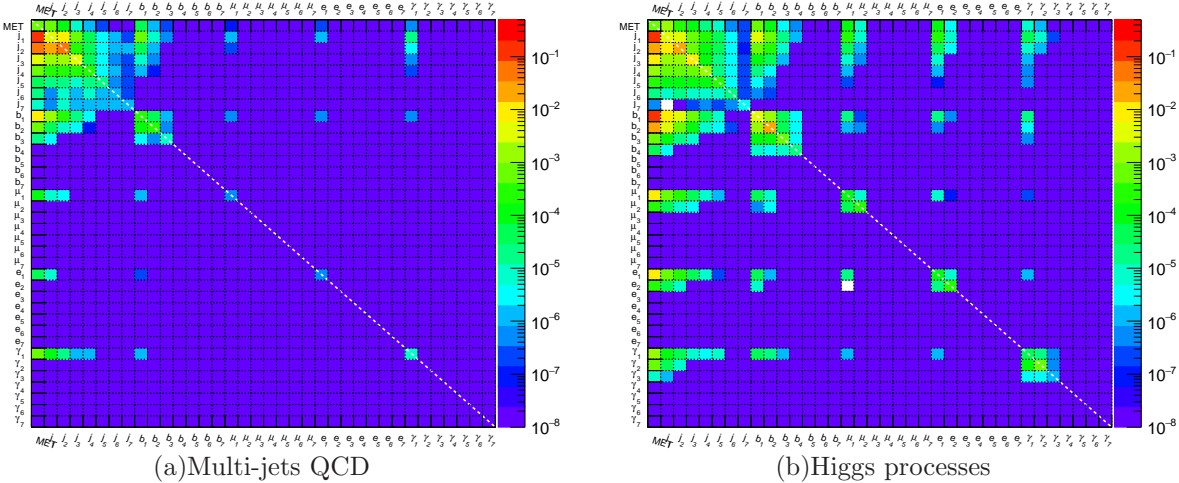

FIG. 1. Visualization of the RMMs for (a) multijet QCD events and (b) for the Standard Model Higgs production (all decays of the Higgs are allowed). The definition of the RMM is given in the text. The RMMs were obtained after averaging over 100,000 $pp$ collision events generated using Pythia8 after parton showers and hadronization.

jets with the muon ID in 1% cases. The fake rates considered here are representative and, generally, are the upper limits for the rates discussed in [11]. The minimum value of the transverse momentum of all leptons and photons was 20 GeV. The missing transverse energy is recorded above 50 GeV.

To prepare the event samples for an ANN event classification, the events were transformed to the RMMs with five types ($T = 5$) of the reconstructed objects: jets ($j$), $b$-jets ($b$), muons ($\mu$), electrons ($e$) and photons ($\gamma$). Up to seven particles per type were considered ($N = 7$), leading to the so-called T5N7 topology for the RMM inputs. This transformation creates the RMMs with a size of 36×36. Only nonzero elements of such sparse matrices (and their indexes) are stored for further processing. Figure 1 shows the RMMs for multijets QCD events and the SM Higgs production after averaging the RMMs over 100,000 $pp$ simulated collision events. As expected, differences between these two processes seen in Figure 1 are due to the decays of the SM Higgs boson.

To illustrate the event-classification capabilities using the common RMM input space for different event categories, we have chosen to use a simple shallow (with one hidden layer) backpropagation ANN from the FANN package [12]. If the classification works for such

a simple and well-established algorithm, this will build a baseline for future exploration of more complex machine-learning techniques. The sigmoid activation function was used for all ANN nodes. No re-scaling of the input values was applied since the range $[0, 1]$ is fixed by the RMM definition. The ANN had 1296 input nodes mapped to the cells of the RMM, after converting the matrices to one-dimensional arrays. A single hidden layer had 200 nodes, while the output layer had five nodes, $O_i$, $i = 1, \ldots 5$, corresponding to five types of events. Each node $O_i$ of the output layer was assigned the value 0 ("false") or 1 ("true") during the training process. The QCD multijet events correspond to $O_1 = 1$ (with all other values being zero), the Standard Model Higgs events correspond to $O_2 = 1$ (with all other values being zero) and so on. According to this definition, the value $O_i$ of the output node corresponds to the probability for identification of a given process.

The goal of the ANN training was to reproduce the five values of the output layer for the known event types. During the training, a second ("validation") data sample was used, which was constructed from 20,000 RMMs for each event type. Figure 2 shows the mean squared error (MSE) as a function of the number of epochs during the training procedure. The dashed line shows the MSE for the independent validation sample. As expected for a well-behaved training procedure, MSE values decrease as the number of epochs increases. The effect of over-training was observed after 100 epochs, after which the validation dataset did not show a decreasing trend for the MSE errors. Therefore, the training was terminated after this number of epochs. After the training, the MSE decreases from 0.8 to 0.065. (The value of 0.4 corresponds to the case when no training is possible). It is quite remarkable that the training based on the RMM converges after the relatively small number of epochs[1].

The trained ANN was applied to a third independent sample with 100,000 RMMs from all five event categories. Figure 3 shows the values of the output layer of the trained network for the charged Higgs, SM Higgs, $t\bar{t}$ and double $W/Z$ production. The ANN output values for multijet QCD events are not shown to avoid redundancy in presenting the results. As expected for robust event identification, peaks near 1 are observed for the four considered process types.

The success of the ANN training was evaluated in terms of the purity of identified events at a given value on the ANN output node. This purity was defined as a ratio of the number

---

[1] The RMM+ANN training with 100 epochs took 3 hours using 16 threads of the Intel Xeon E5-2650 (2.20GHz) CPU.

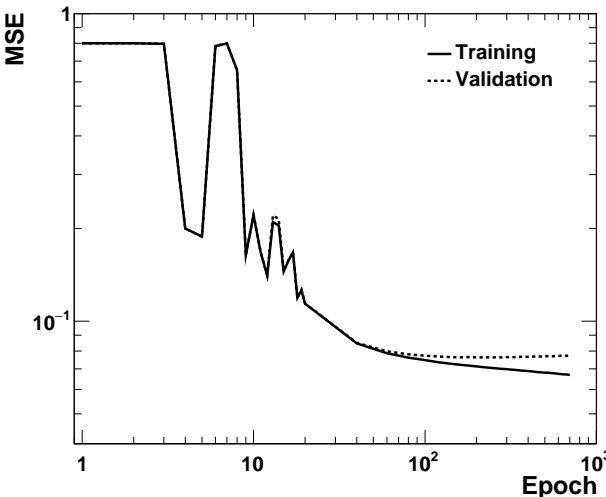

FIG. 2. Variation of mean squared error (MSE) vs the number of epochs for training and validation RMM data.

of events passed a cut of 0.8 on the ANN output score for a given input, divided by the total number of accepted events (regardless of their origin) above this cut. The purity of events for $t\bar{t}$, $H^+$ and SM Higgs was close to 90%, while the purity for the reconstruction of the double-boson process was 80%. The dominant contributor to the background in the latter case was the $t\bar{t}$ process.

## III. BACKGROUND REDUCTION FOR BSM SEARCHES

One immediate application of the RMM is to reduce a large rate of background events from SM events and to increase signal-over-background (S/B) ratios for exotic processes. As discussed before, the RMMs can be used as generic inputs without handcrafting variables for each SM and BSM event type. As a result, a single neural network with unified input feature space and multiple output nodes can be used.

In this example, we will use MC events that, typically, have at least one lepton and two hadronic jets per event. The jets can be associated with decays of heavy resonances. The Pythia Monte Carlo model was used to simulate the following event samples:

- Multijet QCD events preselected with at least one isolated lepton. The lepton isolation was discussed in Sect. II. In order to increase the statistics in the tail of the jet

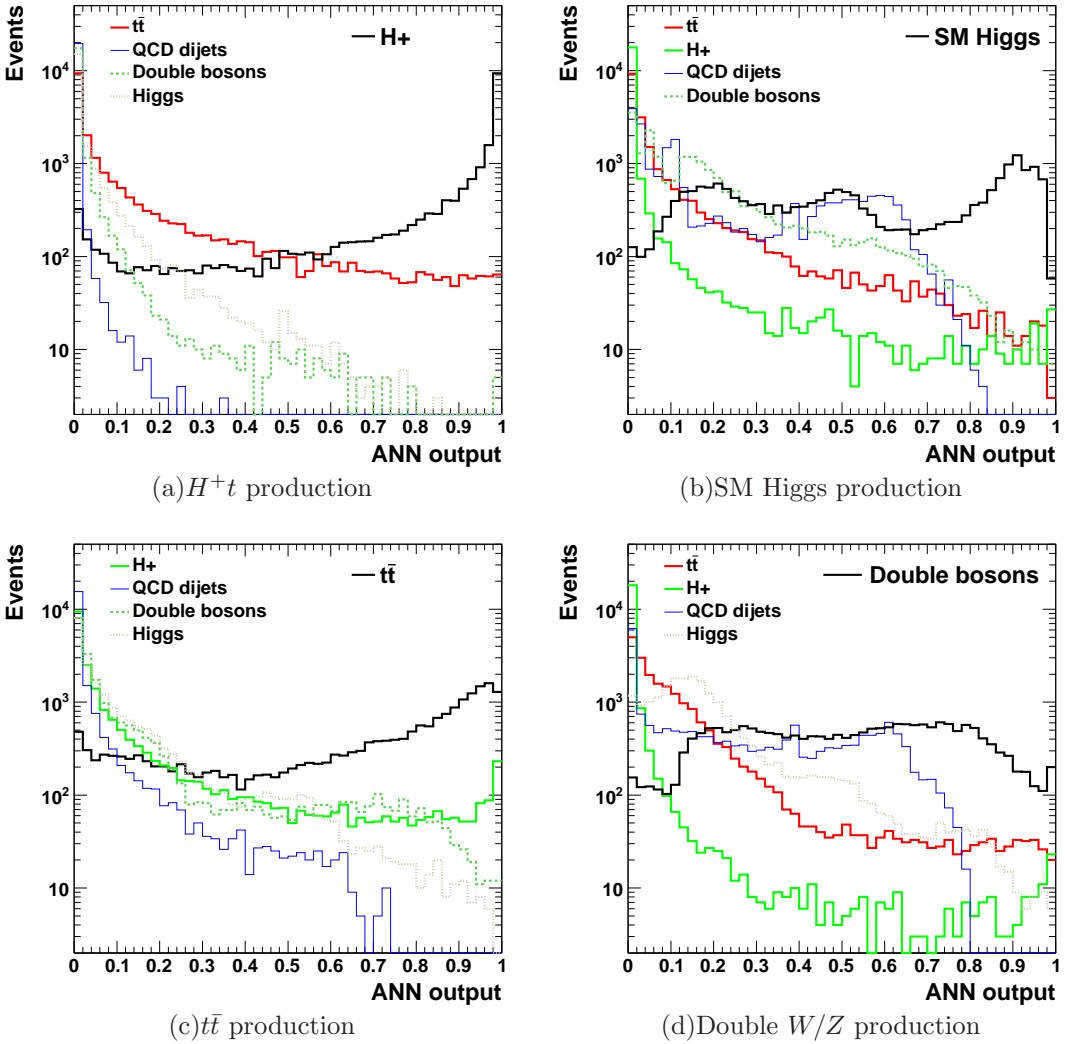

FIG. 3. The ANN output scores for Monte Carlo events with the charged Higgs ($H^+t$), SM Higgs, $t\bar{t}$ and double $W/Z$ production. The black solid lines show the output for a given (known) event category. Other lines show the values of other nodes in the output ANN layer. The $pp$ collision events were generated using Pythia8 after parton showers and hadronization. All decays of heavy bosons and top quarks allowed by the Pythia8 were included.

transverse momentum distribution, we apply a phase-space re-weighting technique [3] for $2 \to 2$ all QCD processes;

- Standard Model $W$+jet, $Z$+jets, Higgs, $t\bar{t}$ and single-top events combined according to their corresponding cross sections. This event sample has a large rate of events with leptons and jets, thus it should represent the major background for BSM models

predicting high rates of leptons and jets;

- $H^+t$ events as discussed in Sect. II;

- $Z'/W'$ events from the Sequential Standard Model (SSM) [13]. In this BSM model, $W' \to WZ'$, where $W$ decays leptonically into $l\nu$ and the heavy $Z'$ decays hadronically into two jets;

- A $\rho_T$-model. It is a variation of technicolor models [14] where a resonance, $\rho_T$, decays through the $s$-channel to the SM $W$ boson and a technipion, $\pi_T$, where the $W$ and $\pi_T$ subsequently decay into leptons and jets, respectively;

- A model with heavy $Z'$ from the process $q\bar{q} \to Z'W$ in a simplified Dark Matter model [15] with the $W$ production, where a $Z'$ decays to two jets, while $W$ decays leptonically into $l\nu$.

In order to create a SM "background" sample for the BSM models considered above, the first and the second event samples were combined together using the cross sections predicted by Pythia. The event rates of the latter four BSM models, defined as $H^+t$, $Z'/W'$ SSM, $\rho_T$ and $Z'$ (DM), are also predicted by this MC generator, with the settings given in [7]. The $\rho_T$ and $Z'/W'$ SSM models and their settings were also discussed in [16]. The BSM models were generated assuming 1, 2 and 3 TeV masses for the $Z'$, $\rho_T$ and $H^+$ heavy particles. About 20,000 events were generated for the BSM models and about 2 million events for the SM processes. For each event category, all available sub-processes were simulated at leading-order QCD with parton showers and hadronization. Jets, $b$-jets, isolated electrons and muons and photons were reconstructed using truth-level information as described in Sect. II. The minimum transverse momentum of all leptons was set to 30 GeV, while the minimum transverse momentum of jets was 20 GeV.

Note that all six processes considered above have similar final states since they contain a lepton and a few jets. Therefore, separation of such events is somewhat more challenging for the RMM+ANN, compared to the processes discussed in Sect. II, which had distinct final states (QCD dijet events were not preselected by requiring an identified lepton).

Similar to Sect. II, the generated collision events were transformed to the RMM representation with five types ($T = 5$) of the reconstructed objects. The capacity of the matrix was increased from $N = 7$ to $N = 10$ in order to allow contributions from particles (jets)

with small transverse momenta. This "T5N10" input configuration leads to sparse matrices of the size $51 \times 51$.

The T5N10 RMM matrices were used as the input for a shallow backpropogation neural network with $51 \times 51 = 2601$ input nodes. The ANN had a similar structure as that discussed in Sect. II: A single hidden layer had 200 nodes, while the output layer had six nodes, $O_i$, $i = 1, \ldots 6$, corresponding to six types of the considered events. Each node $O_i$ of the output layer has the value 0 ("true") or 1 ("false"). The ANN contained 2809 neurons with 521606 connections. The training was stopped after 200 epochs after using an independent validation sample. The CPU time required for the ANN training was similar to that discussed in the previous section.

For each MC sample, dijet invariant masses, $M_{jj}$, were reconstructed by combining the two leading jets having the highest values of jet transverse momentum. The $M_{jj}$ variable will be the primary observable for which the impact of the ANN training procedure will be tested. To avoid biases for the $M_{jj}$ distribution after the application of the ANN, all cells associated with the $M_{jj}$ variables were removed from the ANN training. The following cell positions were set to contain zero values:

- (1,1) which corresponds to the energy of a leading jet;

- (1+$N$,1+$N$) which corresponds to the energy of a leading $b$-jet;

- (2,1) which corresponds to the $M_{jj}$ of two leading light-flavor jets;

- (1+$N$,2+$N$) which corresponds to the $M_{jj}$ of two leading $b$-jets;

- (1+$N$,2) which corresponds to the $M_{jj}$ of one leading light-flavor jet and $b$-jet.

Figure 4 shows the values of the output neurons that correspond to the four BSM models, together with the values for the SM background processes. As expected, the ANN outputs are close to zero for the SM background, indicating a good separation power between the BSM models and the SM processes in the output space of the ANN. According to this figure, background events can be efficiently removed after requiring the output values on the BSM nodes above 0.2.

Figures 5 show the $M_{jj}$ distributions for the background and signal events for $pp$ collisions at $\sqrt{s}$=13 TeV using an integrated luminosity of 150 fb$^{-1}$. The SM background was a sum of

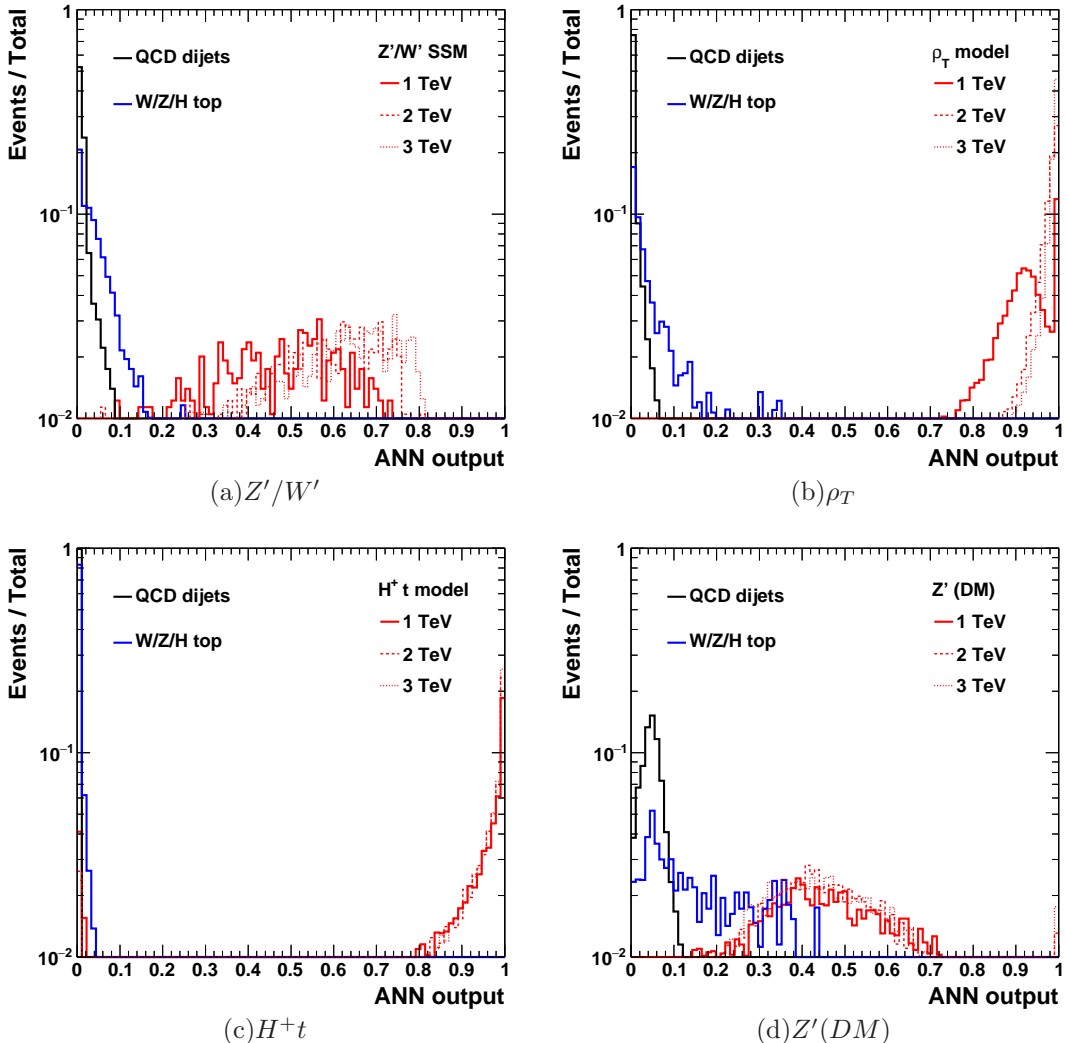

FIG. 4. The ANN output scores for the output nodes of the BSM models for the background (thick black and blue solid lines) and the corresponding to this node BSM model for different mass assumptions (the red lines). The $pp$ collision events were generated using Pythia8 after parton showers and hadronization.

the dijet QCD sample and the sample that includes SM $W/Z/H$ and top events. The BSM signals discussed above are shown for three representative masses, 1, 2 and 3 TeV, of $Z'$, $\rho_T$ and $H^+$ heavy particles. The first two particles decay into two jets, giving rise to peaks in the $M_{jj}$ distributions at similar masses. The $H^+$ boson has more complex decays ($tb$) with multiple jets, but two leading jets still show broad enhancements near (but somewhat below) 1, 2 and 3 TeV masses.

Figures 5(b),(e),(h),(k) show the dijet masses after accepting events which have values

for the BSM output nodes above 0.2. In all cases the ANN increases the S/B ratio after applying the ANN-based selection. The SM background was reduced by several orders of magnitudes, while the signal was decreased by less than 20%. The actual S/B values, as well as the other plots shown in Fig. 5, will be discussed in the next section.

## IV.    MODEL-INDEPENDENT SEARCHED FOR BSM SIGNALS

Another interesting application of the RMM is to perform a model-agnostic survey of the LHC data, or creating an event sample that does not belong to the known SM processes. In our new example, the goal will be to improve the chances of detecting new particles after rejecting events triggered as being SM event types, assuming that the ANN training is performed without using the BSM events. In this sense, the trained ANN will represent a "fingerprint" of kinematics of the SM events. This model-independent (or "agnostic") ANN selection is particularly interesting since does not require Monte Carlo modeling of BSM physics.

For this purpose, we use the T4N10 RMM with an ANN that has two output nodes: one output corresponds to the dijets QCD events, while the second output corresponds to an event sample with combined $W$+jet, $Z$+jets, SM Higgs, $t\bar{t}$ and single-top events. This ANN with 2805 nodes had 520802 connections that need to be trained using the RMM matrices constructed from the SM events.

The ANN was trained with the T4N10 RMM inputs, and then the trained ANN was used as a filter to remove the SM events. The MSE values were reduced from 0.5 to 0.025 after 200 epochs during the training process. The ANN scores on the output nodes had well-defined peaks at 1 for the nodes that correspond to each of the two SM processes. In order to filter out the SM events, the two output neurons associated with the SM event samples were required to have values below 0.8. The result of this procedure is shown in Fig. 5 (c),(f),(i),(l). It can be seen that the SM background contributions are reduced, without visible distortions of the $M_{jj}$ distributions.

Figure 6 shows the values of the S/B ratios for the BSM models shown in Fig. 5. Such ratios were defined by dividing the numbers of events of the BSM signal distributions by the number of events in the SM background near the mass regions with the largest numbers of BSM events of the $M_{jj}$ distributions. The widths of the regions around the peak positions,

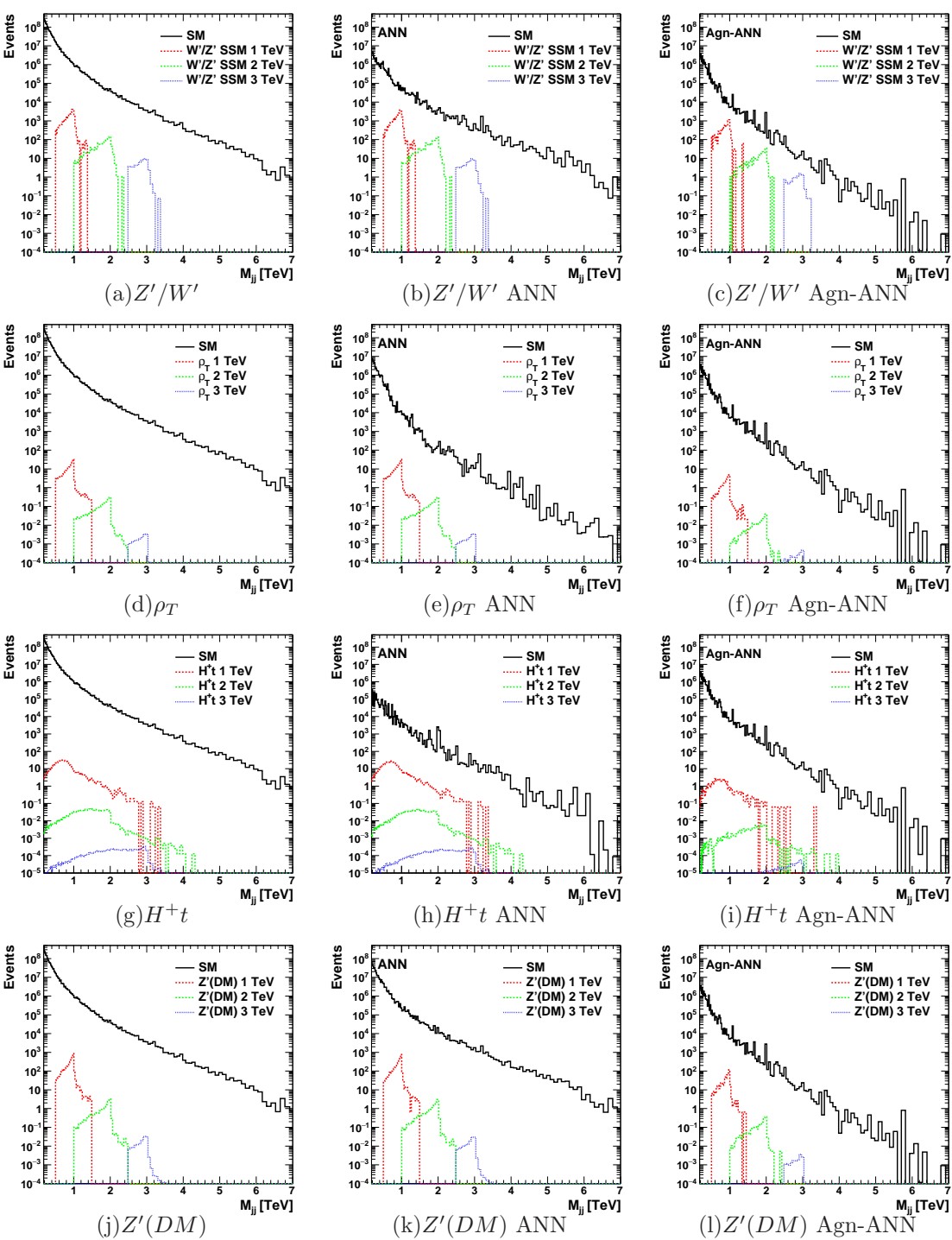

FIG. 5. The $M_{jj}$ distributions for 150 fb$^{-1}$ of $pp$ collision at $\sqrt{s} = 13$ TeV for the background and BSM models (shown for three representative masses of heavy particles) before and after applying the RMM+ANN selections. (b),(e),(h),(k) use a cut on the ANN scores for the BSM output nodes. (c),(f),(i),(l) show a model-independent selection that does not use BSM events for the ANN training.

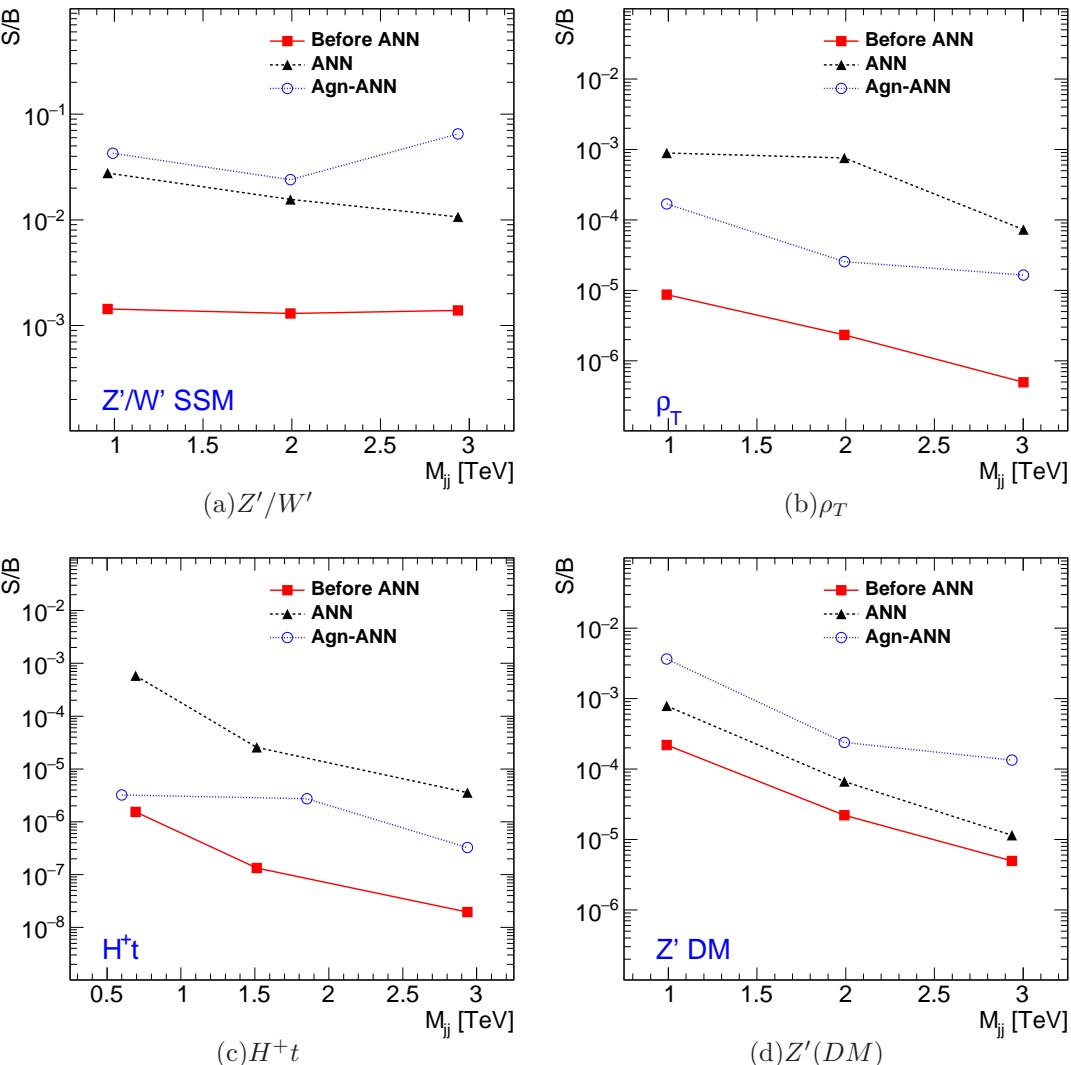

FIG. 6. The values of the signal-over-background (S/B) ratios for three masses of BSM heavy particles that create $M_{jj}$ peaks. The values are shown for the original $M_{jj}$ distribution, after applying cuts on the output nodes of the corresponding BSM event types (labeled as "ANN"), and for a model-independent (agnostic) selection (labeled as "Agn-ANN") without using the BSM events for the ANN training. The latter S/B ratios were obtained by filtering out the SM processes after requiring that the SM output nodes should have values smaller than 0.8.

where the events were counted, correspond to the root mean square of the $M_{jj}$ histograms for the BSM models. The S/B ratios are shown for the original $M_{jj}$ distribution before applying the ANN, after applying the RMM+ANN assuming that the output ANN scores are larger than 0.2 on the output node that corresponds to the given BSM process (labeled

as "ANN") and for the model-independent RMM+ANN ("agnostic") selection (the column "Agn-ANN"). The latter S/B ratios were obtained by removing the SM events after requiring that the output scores on the SM nodes should have values smaller than 0.8, after using RMM+ANN training without the BSM events.

This example shows that the S/B ratio can be increased by the RMM+ANN by a factor 10-500, depending on the masses and process types. It can be seen the S/B ratios for the $\rho_T$ and $H^+t$ models have a larger increase for BSM-specific selection ("ANN"), compared to the BSM-agnostic selection ("Agn-ANN"). The other two BSM models show that the model-agnostic selection can even over-perform the BSM specific selection.

The latter observation is rather important. It shows that the RMM+ANN method can be used for designing model-independent searches for BSM particle without the knowledge on specific BSM models. In the training procedure, the neural network "learns" the kinematic of identified particles and jets from SM events produced by Monte Carlo simulations. Then the trained ANN can be applied to experimental data to create a sample of events that is distinct from the SM processes, i.e. may contain potential signals from new physics. The ANN trained using the SM events, in fact, represents a numeric filter with kinematic characteristics of the SM expressed in terms of the trained neuron connections after using the RMM inputs.

A few additional comments should follow:

- The removal of the cells associated with the $M_{jj}$ may not be required for model-agnostic searches since the current procedure does not use BSM models with specific masses of heavy particles. When such cells are not removed, the S/B ratio was increased compared to the case when the cell removal was applied;

- The observed increase in the S/B ratio was obtained for processes that have already significant similarities in the final states since they include leptons and jets. If no lepton selection is applied to the multi-jet QCD sample, the S/B ratio will show a larger improvement compared to what is shown in Fig. 6;

- For simplicity, this study combines the $W$+jet, $Z$+jets, Higgs, $t\bar{t}$ and single-top events into a single event sample with a single output ANN node. The performance of the ANN is expected to be better when each distinct physics process is associated with its own output node since more neutron connections will be involved in the training;

- As a cross-check, an ANN with two hidden layers, with 300 and 150 nodes in each, was studied. Such a "deep" neural network had 3056 neurons with 826052 connections (and 3060 neurons and 826656 connections in the case with six outputs). The training took more time than in the case of the three-layer "shallow" ANN discussed above. After the termination of the training of the four-layer ANN using a validation sample, no improvement for the S/B ratio was observed compared to the three-layer network.

Model-independent searches using convolutional neural networks (CNN) applied to backgrounds only was discussed in [17] in the context of jet "images". The approach based in the RMM+ANN does not directly deal with jet shape and sub-structure variables since they are not a part of the standard RMM formalism. The RMM+ANN covers the most important kinematic features of all identified particles and jets and, generally, does not require CNNs which are known for being not easy to train.

## V. QCD DIJET CHALLENGE

A more challenging task is to classify processes that have a mild difference between their final states. As an example of such processes, we will consider the following two event types: $gg \rightarrow gg$ and $qg \rightarrow qg$. Unlike the processes discussed in the previous sections, the final state consists of two jets from the hard LO process and a number of jets from the parton shower followed by hadronization. The event signatures of these two SM processes are nearly identical in terms of particle composition. Perhaps the best-known variable for separation of gluon and quark initiated jets is the number of jet constituents. This number is larger for gluon-initiated jets due to a larger gluon color factor ($C_A = 3$) compared to the quark color factor ($C_F = 4/3$). This can be seen in Fig. 7(a). Therefore, we will use the number of jet constituents of leading and sub-leading in $p_T$ jets for ANN training. Note that jet shape and jet substructure variables can also be used (see, for example, [18]), but we limit our choice to the number of jet constituents which are outside the standard definition of the RMM.

The presence of an extra gluon in the process $gg$ compared to $qg$ leads to small modifications of some event characteristics. For example, Figs. 7(b)-(c) show the distributions of the number of jets per event and the transverse momentum of leading in $p_T$ jets. None of the analyzed kinematic distributions indicate significant differences between $qg$ and $qg$ so that these processes can easily be separated using cut-and-count methods.

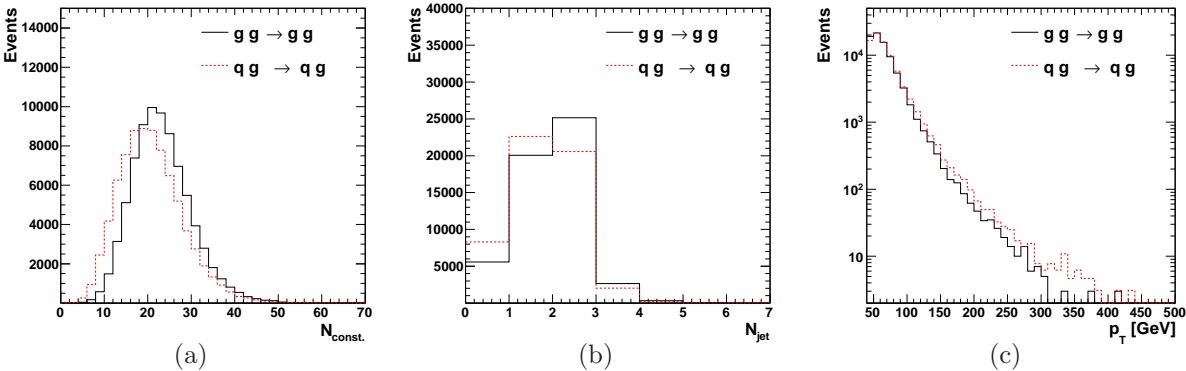

FIG. 7. The comparison between several distributions for $gg \to gg$ and $qg \to qg$ processes. The $pp$ collision events were generated using Pythia8 after parton showers followed by hadronization. (a) The distributions of the number of jet constituents (for all jets above $p_T = 40$ GeV). (b) The distribution of the numbers of jets. (c) The transverse momentum of leading in $p_T$ jets.

The standard procedure for $gg$ and $qg$ event separation is to handpick variables with expected sensitivity to differences between jets initiated by quarks and gluons. Generally, a guiding principle for defining the feature space in such cases does not exist. In addition to the number of jet constituents for two leading jets, the following five input variables were selected: the total number of jets above the $p_T = 40$ GeV, jet transverse momentum, rapidity and the number of constituents of two leading jets, which also show some sensitivity to the presence of the gluon shown in Fig. 7(b)-(c). Since the ANN variables need to be defined using some arbitrary criteria, we will call this approach "pick-and-use" (PaU). Thus the final ANN consists of 7 input nodes, 5 hidden nodes, and one output node, with zero value for the $gg$ process and one for $qg$.

In the case of the RMM approach, instead of the five variables from the PaU method, we used the standard RMM discussed in the previous sections. Thus the input had the RMM plus the number of jet constituents (scaled to the range [0,1]). This leads to 36×36+2 input nodes. The output contained one node with the value 1 for $gg$ and 0 for $qg$ events.

The ANN training was stopped after using a control sample. The results of the trained ANN is shown in Fig. 8(a). One can see that $gg$ and $qg$ processes can be separated using a cut at around 0.5 on the ANN output. The separation power for the PaU and RMM is similar but not the same: The separation between $gg$ and $qg$ is for the RMM is better than for the PaU method. The PaU method leads to the purity of 65% for identification of the

*gg* process after accepting events with the output node values larger than 0.5. The selection purity is 68% for the RMM inputs. The main gain of the latter approach is in the fact that the RMMs simplify the usage of machine learning, eliminating both a time-consuming feature-space study, as well as sources of ambiguity in preparing the input variables.

It is important to note that the standard RMM input can bring rather unexpected improvements for event classifications that can easily escape attention in the case of a hand-crafted input for machine learning. For example, *qg* has a larger rate of isolated photons radiated of the quark from the hard process (this can be found by analyzing the RMM images). This leads to an additional separation power for the RMM inputs. In contrast, the PaU approach relies on certain expectations. In the case of the complex final states with multiple decay channels considered in the previous sections, the identification of appropriate ANN feature space becomes a complex task with the detrimental effects of ambiguity.

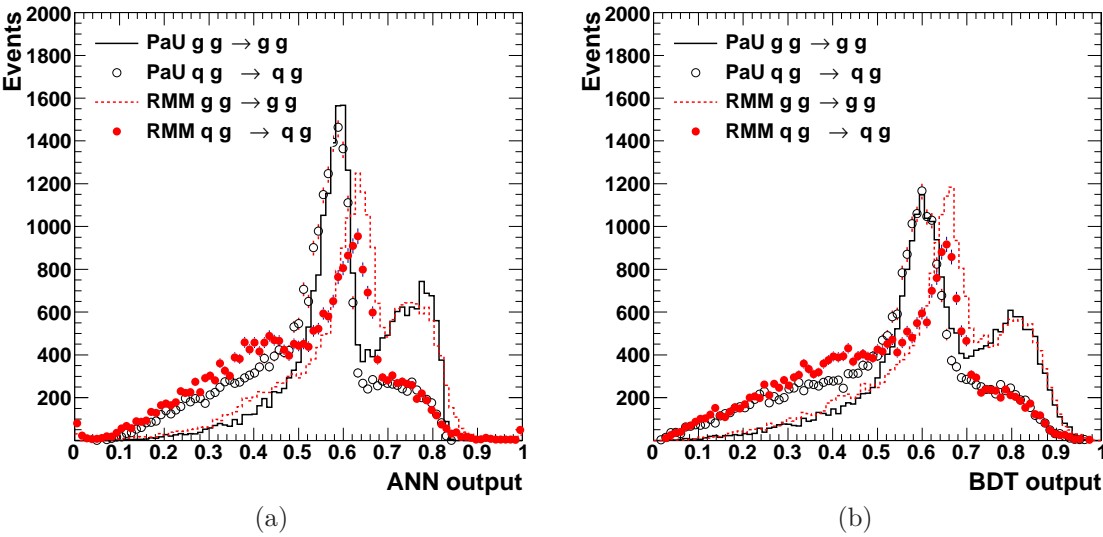

FIG. 8. (a) The ANN output for *gg* and *qg* events in the case of "pick-and-use" (PaU) approach based on seven selected input variables (see the text) and the RMM. (b) The output of boosted decision tree (BDT) for the same inputs as in (a). The events were generated using Pythia8 after parton showers followed by hadronization.

It should be pointed out that the choice of the ANN architecture with the RMM is left to the analyzer. As a check, in addition to the backpropogation neural network, we also considered a stochastic gradient boosted decision tree with the PaU and RMM variables. The boosted decision tree (BDT) was implemented using the FastBDT package [19]. The

BDT approach used 100 trees with a depth of 5. Fig. 8(b), confirms that the separation of $gg$ from $qg$ is more effective for the RMM inputs. However, the overall separation was found to be somewhat smaller for the BDT compared to the ANN method.

## VI.  CONCLUSION

The studies presented in this paper demonstrate that the RMM transformation for supervised machine learning provides an effective framework for general event classification problems. In particle physics, machine learning algorithms typically use handcrafted subject-specific variables. Such variables can be replaced by the standard RMMs which are sensitive to a broad class of final-state phenomena in particle collisions by design. As a result, tedious engineering of ANN input space for different event types can be automated. At the same time, wide and shallow neural networks with multiple output nodes can be used. This paper demonstrates a few such applications using the simplest backpropogation ANN and BDT techniques. It was shown that such algorithms are fast and well convergent during the training. Among several examples presented in this paper, the model-independent search that uses the ANNs trained on SM backgrounds only is an interesting direction towards model-independent BSM searches.

### ACKNOWLEDGMENTS

I would like to thank Dr. V. Pascuzzi, Dr. A. Milic, W. Islam and H. Meng for providing Monte Carlo setting for BSM models used in this paper. I especially thank Dr. V. Pascuzzi for discussions of the machine learning methods used in this paper. The submitted manuscript has been created by UChicago Argonne, LLC, Operator of Argonne National Laboratory (Argonne). Argonne, a U.S. Department of Energy Office of Science laboratory, is operated under Contract No. DE-AC02-06CH11357. The U.S. Government retains for itself, and others acting on its behalf, a paid-up nonexclusive, irrevocable worldwide license in said article to reproduce, prepare derivative works, distribute copies to the public, and perform publicly and display publicly, by or on behalf of the Government. The Department of Energy will provide public access to these results of federally sponsored research in accordance with the DOE Public Access Plan.

http://energy.gov/downloads/doe-public-access-plan. Argonne National Laboratorys work was funded by the U.S. Department of Energy, Office of High Energy Physics under contract DE-AC02-06CH11357.

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
