# Peer review of "Machine learning using rapidity-mass matrices for event classification problems in HEP"

_SciPost Physics_

## Round 3 · Referee Report · Anonymous · 2019-10-15

Strengths

1) The manuscript shows the application of the "rapidity-mass matrix" (RMM) to several classification problems at the LHC. In particular, these are the discrimination of charged Higgs-boson, heavy gauge-boson and technipion production from the main backgrounds, a study on anomaly detection and the separation of quark+gluon events from diquark events.
2) The proposed automatic feature engineering with the RMM may be useful in cases where a shallow neural network is trained, for example if only a small number of events is available for training.
3) The proposed method of RMMs is also useful for visualization purposes due to its compact matrix format, although the classification methods do not make use of the format and rather transform the matrix to a one-dimensional array.

Weaknesses

1) The practical usability of the proposed classification strategies is not clearly demonstrated and it mostly ignores developments in applications of deep learning in high-energy physics (there is only a reference to a CNN study, i.e. Ref. [17]). The approaches studied in this manuscript should be put in context of state-of-the-art machine learning studies, in particular deep learning, and the strengths of the proposed approach should be clearly demonstrated (by comparing to such methods as state-of-the-art benchmarks) or by making a compelling argument for the usefulness of the proposed approach over such methods (focussing on the usefulness of RMM for training in low-statistics scenarios may be such an argument). For the three applications studied in this manuscript, I suggest to consider in particular the following advancements: (a) BSM signatures: examples of event classifiers for BSM signatures can be found in the review in 1806.11484 (Ann. Rev. Nucl. Part. Sci. 68 (2018) 161), (b) anomaly detection: auto-encoders, for example 1807.10261 and 1808.08992, (c) quark-gluon tagging: please refer again to the review (1806.11484), but also to newer work, such as 1812.09223 (SciPost Phys. 6 (2019) 069).
2) As the manuscript disregards advancements in deep learning, several statements need to be corrected: On p.2 in the last paragraph, the author mentions "repetitive and tedious tasks of feature-space engineering", which are obsolete if deep learning on large datasets is performed. On p.5 on the top of the page, the author states "this will build a baseline for future exploration of more complex machine-learning techniques", however it is unclear if an improved feature-engineering will help to advance such techniques, as many of them now rely on deep learning. On p.5 in the last sentence of the 2nd paragraph, the author states "It is quite remarkable that the training based on the RMM converges after the relatively small number of epochs." This statement should be demonstrated to be true by a comparison to another benchmark algorithm. In the last paragraph of Section IV, the author compares to a CNN for model-independent searches. A comparison to auto-encoders (see above) would be more suitable, though. On page 18, in the 2nd sentence of the conclusions, the statement "typically use handcrafted subject-specific variables" needs to be revised.
3) The introduction fails to properly introduce the concept of the RMM and merely refers to Ref. [1] and then discusses some features of the RMM. I have to say that Ref. [1] is well written and clearly explains the RMM, but without reading large parts of Ref. [1] in detail, it is not possible to understand the concept of the RMM from this paper alone. I highly suggest to modify the introduction by introducing the concept more carefully, at least give the definition of the RMM and discuss the entries of the matrix in a compact way. In addition, I recommend to remove conclusions (first two sentences in the introduction's third paragraph) from the introduction.
4) While the studies in Sections III and IV (classification of BSM signals) are well motivated, the study in Section V ("QCD dijet challenge") is not as motivated. The goal is to distinguish gg->gg vs. gq->gq events, hence alluding to the concept of quark-gluon tagging, which - however - does not operate on event basis but on object (jet) basis. The author also cites Ref. [18] as an example of such a jet-based quark-gluon tagging. It is unclear if the proposed event classifier can be used in experiments. While the study shows that it is possible to distinguish gg from qg events, it is not clear if features that distinguish quark from gluon jets are exploited, or whether the differences aren't rather due to other effects, such as the difference in quark and gluon parton distribution functions. I urge the author to make a more convincing case for the approach that is presented or to remove this study from the manuscript.

Report

The manuscript shows the application of the "rapidity-mass matrix" (RMM) to several classification problems at the LHC. In particular, these are the discrimination of charged Higgs-boson, heavy gauge-boson and technipion production from the main backgrounds, a study on anomaly detection and the separation of quark+gluon events from diquark events. The proposed automatic feature engineering with the RMM may be useful in cases where a shallow neural network is trained, for example if only a small number of events is available for training. The proposed method of RMMs is also useful for visualization purposes due to its compact matrix format, although the classification methods do not make use of the format and rather transform the matrix to a one-dimensional array.

However, the practical usability of the proposed classification strategies is not clearly demonstrated and it mostly ignores developments in applications of deep learning in high-energy physics (there is only a reference to a CNN study, i.e. Ref. [17]). The approaches studied in this manuscript should be put in context of state-of-the-art machine learning studies, in particular deep learning, and the strengths of the proposed approach should be clearly demonstrated (by comparing to such methods as state-of-the-art benchmarks) or by making a compelling argument for the usefulness of the proposed approach over such methods (focussing on the usefulness of RMM for training in low-statistics scenarios may be such an argument). For the three applications studied in this manuscript, I suggest to consider in particular the following advancements: (1) BSM signatures: examples of event classifiers for BSM signatures can be found in the review in 1806.11484 (Ann. Rev. Nucl. Part. Sci. 68 (2018) 161), (2) anomaly detection: auto-encoders, for example 1807.10261 and 1808.08992, (3) quark-gluon tagging: please refer again to the review (1806.11484), but also to newer work, such as 1812.09223 (SciPost Phys. 6 (2019) 069). As the manuscript disregards advancements in deep learning, several statements need to be corrected: On p.2 in the last paragraph, the author mentions "repetitive and tedious tasks of feature-space engineering", which are obsolete if deep learning on large datasets is performed. On p.5 on the top of the page, the author states "this will build a baseline for future exploration of more complex machine-learning techniques", however it is unclear if an improved feature-engineering will help to advance such techniques, as many of them now rely on deep learning. On p.5 in the last sentence of the 2nd paragraph, the author states "It is quite remarkable that the training based on the RMM converges after the relatively small number of epochs." This statement should be demonstrated to be true by a comparison to another benchmark algorithm. In the last paragraph of Section IV, the author compares to a CNN for model-independent searches. A comparison to auto-encoders (see above) would be more suitable, though. On page 18, in the 2nd sentence of the conclusions, the statement "typically use handcrafted subject-specific variables" needs to be revised.

The text is well written and clear throughout. However, the introduction fails to properly introduce the concept of the RMM and merely refers to Ref. [1] and then discusses some features of the RMM. I have to say that Ref. [1] is well written and clearly explains the RMM, but without reading large parts of Ref. [1] in detail, it is not possible to understand the concept of the RMM from this paper alone. I highly suggest to modify the introduction by introducing the concept more carefully, at least give the definition of the RMM and discuss the entries of the matrix in a compact way. In addition, I recommend to remove conclusions (first two sentences in the introduction's third paragraph) from the introduction.

If these concerns can be addressed and the detailed questions and comments listed below can be answered, I recommend publication.

Detailed questions and comments:
1) p.2,l.5: The notion of "popular event signatures of the Standard Model" is vague. Please find a more concrete wording.
2) p.3,l.11: Please specify how the charged Higgs-boson is simulated to decay.
3) p.3,3rd paragraph: You claim that the object reconstruction is the same as in Ref. [1], but I believe that the jet radius parameter of 0.4 is not the same. Please specify this in the text as well as other criteria that might be different.
4) p.3,3rd paragraph: Do you simulate the b-tagging efficiency? How large is it? Please clarify this in the text.
5) p.3,3rd paragraph: The sentence "The minium transverse energy of all jets was 40 GeV in the pseudorapidity range of |eta|<2.5." is amgiuous, because it could mean that only for jets within |eta|<2.5 the pT was required to be > 40 GeV. Please clarify the sentence.
6) p.3,4th paragraph: You discuss lepton isolation and fake rates. Please specify if you have also simulated jet->gamma and electron->gamma fake rates and whether you have considered a photon isolation criterion.
7) p.3,4th paragraph: Please specify the eta cuts for leptons and photons.
8) p.4,figure 1(b): Why do two entries seem to not be filled (white color), while the corresponding entries on the other side of the matrix diagonal seem to be filled?
9) p.4,last paragraph: What are the hyperparameters that come with the FANN package? Are there for example hyperparameters for the optimization step in the training? How did you choose the hyperparameters?
10) p.5,2nd paragraph: Please specify the number of matrices that were used for the training (you mention only the number of matrices used for validation: 20,000).
11) p.5,2nd paragraph: For classification tasks, the cross entropy is typically a good activation function in the output layer. Why did you choose the MSE?
12) p.5,2nd paragraph: I do not understand the statement "The value of 0.4 corresponds to the case when no training is possible." Do you mean that 0.4 corresponds to a random choice? Why does the training then not start at 0.4 but rather at 0.8?
13) p.6,last paragraph of Section II: Please consider adding a confusion matrix to illustrate the cited numbers.
14) p.7,figure 3: It is unclear from the caption and plots how the plots and curves are divided: Do the plots a),b),c),d) correspond to the different output nodes and the distributions are shown for this output node for the different processes? Or is it the other way around, i.e. they correspond to the different process and the different distributions in each plot show the distributions for the different output nodes? I believe that it is the former, but please clarify this in the caption.
15) p.7,figure 3: Please consider discussing some of the features of the distributions, such as the three bumps of the SM Higgs curve in plot (b).
16) p.7,l.2: I believe that "for 2->2 all QCD processes" should be "for all 2->2 QCD processes?
17) p.7,l.3: Please add a description of the W+jets, Z+jets and single-top samples in the MC section earlier.
18) p.7,l.4: Please mention already here that all cross sections are taken from the LO calculation in Pythia.
19) p.8,l.3-5: What are the masses of the W' and Z' bosons? I believe to understand from the next paragraph that the Z' mass is varied from 1 to 3 TeV, but the W' mass is not mentioned. Please clarify in the text.
20) p.8,l.6: What is the mass of the technipion? Please clarify in the text.
21) p.8,1st paragraph: Please specify whether you generate 20,000 (2,000,000) events for each of the BSM models (SM processes) or for all processes together. This is not clear from the text.
22) p.9,l.5: Did you vary the number of nodes in the hidden layer? If yes: How strongly did you vary them and how strongly did the performance change?
23) p.10,figure 4: Please consider discussing why some processes are easier to discriminate from the background than other processes. Please also discuss the distribution of the 1 TeV signal curve in figure (b).
24) p.10,l.3: The text states "The first two particles decay into two jets", but from p.8 I had thought that the rho_T decays to a W boson and a technipion (and the technipion decays to two jets). Please clarify in the text.
25) p.11,l.2: The text claims a reduction of the background by "several orders of magnitudes", while this does not seem to be true from Figures (j) and (k). Please revise the text.
26) p.12,figure 5: What is the origin of the sharp edges of the signal processes. There seem to be no signal events below a certain threshold in M_jj. Is this a problem with the event generation?
27) p.13,figure 6: I wonder whether S/sqrt(B) or some other significance-related variable wouldn't be more appropriate for discussing the improvement in sensitivity to BSM signals.
28) p.14,l.8-9: How can it happen that the model-agnostic selection can over-perform specifically-trained networks? Couldn't this just be an artifact of the arbitrary cuts applied to the NN output?
29) p.14,1st bullet: Wouldn't this bullet also hold for the studies discussed in Section III? It is clear that not removing cells may increase the level of correlation of the NN output with M_jj, but the same arguments holds here. I do not see the distinction in this argument between the model-specific and the model-agnostic searches.
30) p.14,2nd bullet: This statement is rather trivial and I would suggest to remove it. The rejection of QCD multijet events based on requiring a lepton (in an analysis using a 1-lepton selection) does not require machine learning.
31) p.15,l.2: Although it is in quotation marks, I would simply suggest to remove the word "deep" here, because the NN just has two hidden layers.

Requested changes

See report.

---

## Round 3 · Referee Report · Anonymous · 2019-10-24

Report

The author studied a neural network implementation for classifying between different event topolgies, both in the Standard Model and in some Beyond the Standard Model scenarios. The novelty of the approach is in using the Rapidity Mass Matrices to represent the event data. The approach seems to work very well for general event classification problems, and the application of the Rapidity Mass Matrices to these types of problems certainly deserves attention. I do think there are some minor revisions which could really help the reader in better understanding the results and the context of these results in the wider field. Therefore with the following revisions I would recommend that this work be published in SciPost.

Requested changes

1 - In the introduction there should be a short overview of the current state-of-the-art in the application of machine learning techniques to particle physics, such that the context of the work in wider field is clear. For example, there is no mention of the many different top-tagging algorithms that have implemented using machine learning (1902.09914), or the different unsupervised jet classifiers.

2 - In presenting the results for the classification the author should also display the associated ROC curves. For example in Fig. 3 the ROC curves could be plotted for H+ vs all, ttbar vs all, etc. This would be very useful to gain a better understanding of the tagging efficiency and background rejection.

3 - In the discussion for the agnostic tagger, where the neural network learns Standard Model processes and then uses this to reduce backgrounds in BSM searches, the author should perhaps discuss similarities with autoencoder methods (1808.08992). These methods, also based on neural network architectures, learn how to encode and decode Standard Model events such that Standard Model events are reconstructed well in the decoding stage. The reconstruction error in the autoencoder method could have similarities with the failure of the Agn-ANN to classify the event as a particular Standard Model event.

4 - The application to quark vs gluon tagging is very interesting, and as discussed in point 1, it would be good to see the results plotted on ROC curves. It would also be good to see a comparison with other machine learning methods for quark vs gluon tagging in the literature (1612.01551 and 1812.09223). Even just a discussion comparing the performance portryaed through the ROC curves with those presented in the aforementioned papers.

---

## Round 3 · Referee Report · Anonymous · 2019-10-24

Strengths

1- The construction and application of rapidity mass matrix (RMM) is an interesting concept. The elegant feature is that one don't need to design high level variables explicitly but the basic kinematic distribution of all the objects serve the purpose of event categorization.

2- The author shows application in two different context (event classification and QCD dijet challenge).

Weaknesses

1- The nature of the obtained results are not analyzed in details.

Report

In general the manuscript is in good shape. The report is made based on the manuscript : https://arxiv.org/pdf/1810.06669v3.pdf . Please find below the questions to be addressed.

1- Section 2 : When doing the b-jet identification, is it a $\Delta R$ matching between b-quark and the reconstructed jet? Is the applied criteria always ensures that the identified b-jet has a B-hadron whose energy is $\ge$ 50$\%$ of the jet energy? If not, then please quantify the fraction of events for which this correspondence doesn't hold.

2- What were the proportion of five categories of event, used in the training sample. Were the process cross-section or some other variable was used as training weights? If not, can the author please comment on the effect of of training weights on final output?

3- In figure 3c, ANN score for QCD dijet sample, evaluated on SM Higgs sample, has similar profile with SM Higgs score distribution up-to ANN value 0.6. In figure 3d, diboson score and QCD diject score have similar similarity upto a value 0.6. Is it an artifact of any kinematic distribution of the objects in those events or its coming from RMM matrix non-zero cells only?

4- In the ANN training, all the cell associated with $M_{jj}$ were set to zero in order to remove bias. If one scales down $M_{jj}$ by $H_T$ (scalar sum of transverse momenta of all the visible objects in an event, i.e. $\sum p_T$), that should in principle remove the bias. Can the author please cross-check this?

5- In figure 6 : The S/B generally goes down with increasing $M_{jj}$ but in case of $Z'/W'$ sample, Before-ANN is almost flat and Agn-ANN increases at the highest mass point. Can the author please explain the origin of this anomalous behavior.

6- In the case of QCD dijet challenge, is there any change in performance when $qq \rightarrow gg$ method is compared to $gg \rightarrow gg$ ? In the mixed sample of $qg$ and $gg$ events, can teh author please add a ROC curve to show the distinguishing power between these two processes?

7- In general while constructing neural network, can you please add one more hidden layer
and demonstrate that the overtraining starts much earlier?

Requested changes

1- Please add discussion on training weights.
2- Please provide a ROC curve for QCD-dijet challenge.

---

## Editorial Decision

editor-in-charge_assigned